# Planning Strategy to Optimize the Dose-Averaged LET Distribution in Large Pelvic Sarcomas/Chordomas Treated with Carbon-Ion Radiotherapy

**DOI:** 10.3390/cancers15194903

**Published:** 2023-10-09

**Authors:** Ankita Nachankar, Mansure Schafasand, Antonio Carlino, Eugen Hug, Markus Stock, Joanna Góra, Piero Fossati

**Affiliations:** 1MedAustron Ion Therapy Center, 2700 Wiener Neustadt, Austria; mansure.schafasand@medaustron.at (M.S.); antonio.carlino@medaustron.at (A.C.); eugen.hug@medaustron.at (E.H.); markus.stock@medaustron.at (M.S.); joanna.gora@medaustron.at (J.G.); piero.fossati@medaustron.at (P.F.); 2ACMIT Gmbh, 2700 Wiener Neustadt, Austria; 3Department of Radiation Oncology, Medical University of Vienna, 1090 Wien, Austria; 4Division Medical Physics, Karl Landsteiner University of Health Sciences, 3500 Krems an der Donau, Austria; 5Division Radiation Oncology, Karl Landsteiner University of Health Sciences, 3500 Krems an der Donau, Austria

**Keywords:** LETd optimization, distal patching, pelvic sarcoma/chordomas, carbon-ion radiotherapy

## Abstract

**Simple Summary:**

Carbon-ion radiotherapy (CIRT) is a potentially curative treatment for unresectable pelvic sarcomas/chordomas. Tumor control remains suboptimal in large pelvic sarcomas/chordomas compared with the control of small tumors. We hypothesized that lower dose-averaged LET (LETd) distribution in the majority of large tumors could be one of the contributing factors for local relapse. The high-LETd region lies at the end of the beam range. As the tumor size increases, the volume of tumor covered by high LETd decreases. A statistically significant difference was observed in LETd distribution between small and large pelvic sarcomas/chordomas. To improve the high-LETd component in large tumors, LETd optimization using ‘distal patching’ was explored in a planning setting (not implemented clinically). Distal patching significantly increased (a) median LETd in the targets, (b) LETdmin in low-LETd regions of the targets, (c) the GTV fraction receiving LETd of ≥50 keV/µm, and (d) the high-LETd component in the central region of the GTV, without significant compromise in relative biological effectiveness (RBE)-weighted absorbed-dose (D_RBE_) distribution.

**Abstract:**

To improve outcomes in large sarcomas/chordomas treated with CIRT, there has been recent interest in LET optimization. We evaluated 22 pelvic sarcoma/chordoma patients treated with CIRT [large: HD-CTV ≥ 250 cm^3^ (n = 9), small: HD-CTV < 250 cm^3^ (n = 13)], D_RBE|LEM-I_ = 73.6 (70.4–73.6) Gy (RBE)/16 fractions, using the local effect model-I (LEM-I) optimization and modified-microdosimetric kinetic model (mMKM) recomputation. We observed that to improve high-LETd distribution in large tumors, at least 27 cm^3^ (low-LETd region) of HD-CTV should receive LETd of ≥33 keV/µm (*p* < 0.05). Hence, LETd optimization using ‘distal patching’ was explored in a treatment planning setting (not implemented clinically yet). Distal-patching structures were created to stop beams 1–2 cm beyond the HD-PTV-midplane. These plans were reoptimized and D_RBE|LEM-I_, D_RBE|mMKM_, and LETd were recomputed. Distal patching increased (a) LETd50% in HD-CTV (from 38 ± 3.4 keV/µm to 47 ± 8.1 keV/µm), (b) LETdmin in low-LETd regions of the HD-CTV (from 32 ± 2.3 keV/µm to 36.2 ± 3.6 keV/µm), (c) the GTV fraction receiving LETd of ≥50 keV/µm, (from <10% to >50%) and (d) the high-LETd component in the central region of the GTV, without significant compromise in D_RBE_ distribution. However, distal patching is sensitive to setup/range uncertainties, and efforts to ascertain robustness are underway, before routine clinical implementation.

## 1. Introduction

The advantages of carbon-ion beams include both the physical characteristic of Bragg-Peak dose deposition with sharp distal and lateral fall-off, resulting in a sharp penumbra, as well as the biological properties of high relative biological effectiveness (RBE) due to high linear energy transfer (LET) and low oxygen enhancement ratio (OER). These superior biophysical properties of carbon ions translated into better tumor controls in several radioresistant malignancies such as bone and soft tissue sarcomas, non-squamous head and neck cancers, skull base tumors, lung cancers, and gastro-intestinal, genito-urinary, and gynecological malignancies [1,2,3,4,5,6,7,8,9,10,11,12,13,14,15,16,17,18]. The Bragg peak and sharp penumbra of carbon-ion beams enables high dose deposition in the target with minimal normal tissue toxicity [9,10,12,15]. In addition to better local tumor control, CIRT may offer systemic tumor control by activation of several cell cycle signaling and metabolic pathways, and immunogenic cell killing and abscopal effect [19]. The higher LET of carbon ions results in complex, clustered DNA damage which is difficult to repair or often incompletely repaired, resulting in 2.5–3 times higher damage to the DNA compared with that from photons. The efficacy of the high dose-averaged linear energy transfer (LETd) component of CIRT is almost independent from the intrinsic radiosensitivity of tumor cells. Additionally, the oxygen-independent cell killing capacity of carbon ions makes them attractive for selective targeting of radioresistant hypoxic tumors. This is especially relevant in the treatment of sarcomas/chordomas, which are radio-resistant and frequently exhibit a significant intra-tumoral hypoxic component, overall resulting in poor response to low-LETd photon-based radical radiotherapy [20,21,22,23]. Carbon-ion radiotherapy (CIRT) is a curative treatment option for unresectable pelvic sarcomas/chordomas [1,2,3,4,5,6,7,8,9]. It is also used as an alternative to surgery for operable sacral chordomas, both in clinical trials as well as in clinical practice.

One of the important factors influencing outcomes in large pelvic sarcomas/chordomas treated with CIRT is the RBE-weighted absorbed dose (D_RBE_). D_RBE_ depends upon RBE; however, the RBE of carbon ions is variable across the spread of the Bragg peak (SOBP). The spatial distribution of RBE along the beam path depends upon phenomenological and mechanistic RBE models applied for D_RBE_ calculation. Japanese centers employed the mixed-beam model (MBM) for passively scattered CIRT created by Kanai et al. [24,25], which was based on calculations of a linear quadratic model for a passively scattered CIRT beam for 10% survival of human salivary gland cell lines. This model was later substituted by the Microdosimetric Kinetic Model (MKM) for active scanning beams [26,27]. The MKM model used HSG cells as a biological reference system, and D_RBE_ distribution was expressed with respect to the “reference carbon-ion beam”. This model was adapted to match with previous clinical CIRT experience with a MBM model from NIRS (Japan) and called modified MKM (mMKM) [28]. In contrast, European facilities used the local effect model (LEM-I), which was developed by Scholz and Kraft for an idealized cell line with an alpha/beta ratio fixed at 2 Gy [29]. Both LEM-I and mMKM models have their unique advantages, but neither can fully describe the complexity of CIRT efficacy, as they calculate dose which is equivalent to photons for one single end point, disregarding any intrinsic tumor heterogeneity.

Several clinical studies indicate that despite adequate D_RBE_ target coverage according to both the LEM-I and mMKM models, CIRT outcomes in large pelvic sarcomas/chordomas remain suboptimal. Previous reports suggest that local control in large pelvic sarcomas/chordomas is inferior to those obtained in tumors smaller than 200–500 mL [1,2,3,4,5,6,7]. Theoretically, CIRT plans that deliver a high D_RBE_ but not a high enough LETd in a significant portion of the tumor might be the cause for suboptimal tumor control. This has been suggested by some early clinical results in pancreatic tumors, chordomas, and chondrosarcomas [30,31,32]. This has created interest in optimizing LETd distribution to potentially improve outcomes in photon-resistant tumors treated with CIRT [33,34]. The LETd distribution inside the target in a CIRT treatment plans can be affected by a variety of factors, such as tumor volume, shape, D_RBE_ prescription, number and orientation of beams, use of patching structures, optimization algorithm, RBE model employed for optimization, and boost strategies, i.e., sequential vs. simultaneous integrated boost. The quantitative knowledge about the best strategies to optimize high-LETd distribution inside the tumor is still limited. The principal challenge is that the high-LETd region lies at the end of the beam range, and thus at the distal portion of target and even beyond. In contrast, the proximal and central target volumes are treated with lower LETd. Presumably, it is the central tumor component where radioresistant hypoxic cells develop, especially in sarcomas/chordomas.

In this study, we evaluated the differences in the LETd distribution between large and small pelvic sarcomas/chordomas and made an attempt to explore solutions to improve the high-LETd component inside the target while maintaining a reasonably acceptable D_RBE_ distribution.

## 2. Materials and Methods

### 2.1. Patient Characteristics

A retrospective analysis was conducted for 22 patients with pelvic sarcomas/chordomas treated with CIRT at MedAustron Ion therapy Center, Austria between September 2020 and June 2022. Informed consent was obtained from patients for anonymized data analysis and publication of results through an institutional prospective Registry Study (clinicaltrials.gov: NCT03049072 ethics committee: GS1-EK-4/350-2015) and SACRO Trial (clinicaltrials.gov: NCT0298651 ethics committee: GS1-EK-1/189-2019). Patients with histologically confirmed non-metastatic pelvic sarcomas/chordomas, between ages 43–77 years at the time of diagnosis, with performance status 0–1 were included (Table 1). Only 2 patients underwent surgery with R2 resection, and hence received postoperative CIRT. We selected patients based on the hypothesis that in spite of adequate D_RBE_ coverage with both the LEM-I and mMKM systems, patients with larger target volumes have worse tumor control, at least partly because of suboptimal intratumoral LETd distribution. Studies reporting poorer outcomes in larger tumors selected variable criteria to define what is considered a large tumor [1,2,3,4,5,6,7]. Elective CTV volumes/low-dose CTV (LD-CTV) may vary as per contouring guidelines across the different centers. As a consequence, the PTV volumes are also variable. In order to choose a more reliable and consistent clinical reference, we selected a high-dose clinical target volume (HD-CTV) to define the thresholding parameter for small and large tumors. At our center, HD-CTV is defined as GTV + 5 mm geometric margin (anatomically adapted). The small tumors are defined as those with HD-CTV of <250 cm^3^ (n = 9) and large tumors as those with HD-CTV ≥250 cm^3^ (n = 13).

### 2.2. Treatment Simulation and Planning: Clinical

Patients were positioned and immobilized in a prone position in a personalized MOLDCARE^®^ Cushion (polystyrene B) and non-perforated thermoplastic mask. Target volume contouring for 21 patients affected by sacral chordoma was performed as per SACRO trial protocol (ISG-2016-SACRO) [35], and for the sarcoma case as per ASTRO contouring guidelines for axial soft tissue sarcoma [36]. Tumors were treated with CIRT to the LEM-I prescription doses of 73.6 (70.4–76.8) [Gy (RBE)]/16 fractions, (4 fractions/week), LD-PTV was treated with D_RBE|LEM-I|50%_ = 4.4–4.8 [Gy (RBE)] × 10–9 fractions followed by sequential boost to HD-PTV with D_RBE|LEM-I|50%_ = 4.4–4.8 [Gy (RBE)] × 6–7 fractions. There was no dose adjustment applied for clinical CIRT treatment plans as per different treatment volumes. The clinical CIRT plans were optimized using the LEM-I model in the clinical TPS RayStation 8B, 11A and 11B SP1 (RaySearch Laboratorie AB, Stockholm, Sweden). Treatment plans were optimized using multiple field optimization (MFO) using the pencil-beam dose algorithm using the LEM-I model, with alpha and beta values of photons, αγ = 0.1 Gy^−1^ and βγ = 0.05 Gy^−2^, calculated for the transition dose Dt = 30 Gy, for cell nucleus radius rn = 5 μm. Patients were treated at MedAustron, which is a synchrotron-based dual-particle therapy facility (MedAustron Particle Therapy Accelerator MAPTA: non-commercial machine (EBG MedAustron GmbH, Wiener Neustadt, Austria)). This accelerator can deliver clinical carbon ions (C^6+^) in the range of 120–402.8 MeV/n (from 2.9 to 27 cm in water). Fixed horizontal and vertical beam lines were used for CIRT treatment [37]. The typical beam arrangement consisted of at least 2 orthogonal beams, with one vertical and one lateral beam, or two opposed lateral beams and one anterior beam. CIRT treatment plans are treated mainly in an isocentric treatment setup. A non-isocentric treatment setup is used only if the target is proximal to the patient surface and a range shifter needs to be inserted into the beamline [38]. All the cases included in this study were treated with unpatched clinical CIRT plans (patients were not treated with LETd-optimized CIRT plans). Robust optimization against setup uncertainties (±3–5 mm) and range uncertainties (±3.5%) was applied for most of the clinical CIRT plans. The plans were optimized to achieve D_RBE_ conformality with near maximum dose, D_RBE|LEM-I|2%_ < 107% of prescription dose and near minimum dose D_RBE|LEM-I|98%_ > 95% of prescription dose for HD-PTV. As a routine clinical practice, we evaluated HD-CTV dose coverage with the clinical goals of HD-CTV, D_RBE|LEM-I|98%_ > 95% prescribed dose, HD-CTV, D_RBE|LEM-I|2%_ < 107% of prescription dose. In certain cases, critical OARs like recto-sigmoid and small intestines/bowel loops were blocked using avoidance structures in some cases. In cases where OAR constraints could not be met, a coverage of HD-CTV, D_RBE|LEM-I|95%_ > 95% of prescription dose was accepted. Subsequently, doses were recomputed using the mMKM to verify and compare D_RBE_ distribution with the Japanese system of D_RBE_ calculation (Japanese prescription). The mMKM prescription doses adapted as per the translation schema described by Fossati et al. [39]. In-room online patient set-up verification was performed using two orthogonal X-ray acquisitions for bone anatomy registration. At least two re-evaluation CT scans were performed to validate the robustness of the target and OAR dose statistics. 

### 2.3. D_RBE_ and LETd Evaluation

Patient CT scans, structure sets, CIRT plans, CIRT doses, and DICOM files were imported in the research version of TPS RS 11B-SP1 to evaluate D_RBE|LEM-I_, D_RBE|mMKM_, and LETd distribution. Various D_RBE_ and LETd parameters were evaluated for LD-PTV, LD-CTV, HD-PTV, HD-CTV, and GTV, as well as OARs such as rectum, urinary bladder, small intestine/bowel loops, sacral nerves, cauda equina, and skin. Additionally, D_RBE|LEM-I_, D_RBE|mMKM_, and LETd distribution in the 1 and 5 cm shell region beyond LD-PTV (i.e., entrance region dose and LETd) were reported. 

We considered HD-CTV as an important target for tumor control. D_RBE|LEM-I_, D_RBE|mMKM_, and LETd evaluation to HD-CTV is elaborated in the following sections. D_RBE|LEM-I_, D_RBE|mMKM_, and LETd analysis for HD-PTV, GTV, and LETd analysis for OARs is described in Appendix A (figures). For D_RBE|LEM-I_, D_RBE|mMKM_ statistics, D98%, D95%, D50%, and D2% were selected, and for LETd statistics, LETd98%, LETd50%, and LETd2% were selected as relevant datapoints. The conformity index for HD-CTV was calculated by the SALT method CI(SALT) [40] and the homogeneity index was calculated by the Semenenko formula HI(Semenenko) [41] [V(Target): volume of the target, D(prescribed): prescription dose].
(1)$$\mathrm{CI}\left(\mathrm{SALT}\right)=\frac{\;V\left(\mathrm{Target}\right)\;\mathrm{covered}\;\mathrm{by}\;95\%\;\mathrm{of}\;\mathrm{prescribed}\;\mathrm{dose}}{V\left(\mathrm{Target}\right)}$$
(2)$$\mathrm{HI}\left(\mathrm{Semenenko}\right)=\frac{\;D5\%\;-D95\%}{D\left(\mathrm{prescribed}\right)\;}$$

Additionally, the dose volume histogram (DVH) and LETd volume histogram (LVH) for small and large tumors were compared (Figure 1a). Considering the large variation in the target volumes in pelvic sarcomas/chordomas, we decided to focus on the absolute-volume LVH rather than the relative-volume LVH. In order to evaluate the region of the target receiving low LETd distribution, absolute-volume LVHs were created by shifting the volume of HD-CTV of all cases with respect to the largest HD-CTV volume (Figure 1b). The cumulative absolute-volume LVH (mean ± SD) of small and large tumors was compared to assess the volume of the target e.g., HD-CTV receiving the lowest LETd (low-LETd region).

### 2.4. LETd Optimization: Distal Patching

While clinical tools for direct LETd optimization, or optimization of other physical quantities such as physical dose filtered above a given LETd threshold (high-LET-dose), [42,43] were not available in our TPS for clinical use, we explored simple beam modification for LETd optimization using ‘distal patching’ on CIRT plans of previously treated patients with large pelvic sarcomas/chordomas. LETd optimization by distal patching was carried out using the TPS RayStation 11B-SP1. LETd-optimized beam arrangements were designed by introducing ‘distal-patching structures’ (avoidance structures) to original clinical unpatched-CIRT plans to stop beams 1–2 cm beyond the midplane of the PTV in previously treated large pelvic chordomas/sarcomas (Figure 2g–i), allowing 2–4 cm overlap between opposed beams. A new set of plans were created with distal-patching structures which were similar to the original clinical unpatched-CIRT plans, to achieve similar D_RBE|LEM-I_ distribution. Similar to the original clinical unpatched-CIRT plans, robust optimization was applied against setup uncertainties (±3–5 mm) and range uncertainties (±3.5%) for distally patched CIRT plans. CIRT-plans were reoptimized and D_RBE|LEM-I_, D_RBE|mMKM_ and LETd were recomputed. The intention of this exercise was to increase high-LETd distribution inside of GTV and HD-CTV, at the same time maintaining optimal D_RBE_ coverage in these targets. The distal-patching strategy was used only as a planning exercise and was not routinely implemented in the clinic at the time of this study. The optimal geometry of distal-patching structures was decided based on initial experiments on simple geometries such as cubes. We also performed benchmarking of functionalities such as LETd and physical dose filtration based on LET (high-LETd dose) in a commercial TPS against Monte Carlo simulations in an in-silico study [44]. Distal patching allowed redistribution of the high-LETd region from the distal part of PTVs to the central region of GTV and HD-CTV.

Distal patching results in sharper gradients of physical dose with the potential concern of plan robustness with respect to range and setup uncertainties. In order to minimize this issue, distal patching was planned for only 6–7 fractions out of the 16 fractions of treatment. These 6–7 fractions of distally patched treatment represent cone-down boost to HD-PTV. Data analysed for distally patched plans represent statistics from unpatched plans for 10–9 fractions, and from distally patched plans for 6–7 fractions.

### 2.5. Statistical Analysis

Various LETd parameters and absolute LVHs of HD-PTV, HD-CTV, and GTV were compared using either a (two-tailed) *t*-test for normal distribution or a Mann–Whitney U-test for non-normal distributions. Normality was tested with the Shapiro–Wilk test. A *p*-value of <0.05 was considered significant.

## 3. Results

### 3.1. D_RBE|LEM-I_, D_RBE|mMKM_ Evaluation in Small vs. Large Pelvic Sarcomas/Chordomas

In this study, the majority of the patients treated with CIRT had chordoma as histology (21/22), and one had pelvic synovial sarcoma. The mean HD-CTV volume was 116.3 ± 52.6 cm^3^ for small tumors and 551.7 ± 211.3 cm^3^ for large tumors. The average maximum GTV diameter along the beam path was 5.4 ± 2.1 cm for small tumors and 9.1 ± 3.8 cm for large tumors (*p* = 0.01). Patient and tumor characteristics are detailed in Table 1. D_RBE|LEM-I_ and D_RBE|mMKM_ dose statistics for unpatched-CIRT plans were not statistically different between small and large tumors (Table 1).

### 3.2. LETd Evaluation in Small and Large Pelvic Sarcomas/Chordomas

The results of the investigation on D_RBE_ and LETd statistics are presented in Table 1. The average median LETd in HD-CTV for small tumors was 40.2 ± 3.8 keV/µm vs. 38.3 ± 3.2 keV/µm in large tumors. The differences in LETd distribution between small and large tumors is described in the relative-volume LETd volume histogram (LVH) (cumulative LVH, mean ± SD) for HD-CTV (Figure 1a), HD-PTV, and GTV (Appendix A). For this study, the clinical qualitative starting point was that the recurrences may occur partly due to insufficient high-LETd distribution in a significant volume of the tumor. Hence, we further analyzed different regions of the LVH. As seen in Figure 1a, the right-hand part of the LVH showed a moderate-to-high-LETd component concentrated in a very small volume of the HD-CTV in the large tumors. However, a large fraction of the HD-CTV was not covered by high-LETd distribution. Considering the large variation in HD-CTV, evaluating only relative-volume LVH would be suboptimal; hence, we focused on the absolute-volume LVH. In order to evaluate the absolute volume of HD-CTV receiving low-LETd distribution, the left part of the LVH was evaluated. All the HD-CTV volumes were plotted with respect to the largest HD-CTV volume, as described in the methods section (Figure 1b). The difference in LETd distribution between small and large tumors became more prominent on plotting the absolute-volume LVH curves for individual cases. A clear separation is visible between small and large tumor LVHs (Figure 1b). Moreover, a statistically significant difference is demonstrated between the two groups (absolute-volume cumulative LVH representing mean ± SD) for volumes of at least 27 cm^3^ (low-LETd region), for HD-CTV (Figure 1c), and at least 56 and 9 cm^3^ for HD-PTV and GTV, respectively (Appendix A). These volumes represent the low-LETd region of the target volumes and must receive the optimal amount of high-LETd distribution. Our analysis suggests that in order to significantly improve high-LETd distribution in the targets in large sarcomas/chordomas, the low-LETd region should not receive LETd < 33 keV/µm (Figure 1c for HD-CTV and Appendix A for HD-PTV and GTV).

### 3.3. LETd Optimization in Large Pelvic Sarcomas/Chordomas with ‘Distal Patching’

The spatial distribution of carbon-ion spots for unpatched CIRT plan optimized with the LEM-I model is described in a representative case of large pelvic sacral chordoma (Figure 2a–c). The corresponding spatial distribution of D_RBE|LEM-I_, D_RBE|mMKM_, and LETd is described in Figure 2d–f. A slightly higher D2% was noted, for D_RBE|mMKM_ in distally patched plans, of 73.9 ± 3.4 Gy (RBE) (110% of mMKM prescription dose 67.2 Gy (RBE)/ 16 fractions), compared to 71.0 ± 3.6 Gy (RBE) (105.7% of mMKM prescription dose) in unpatched plans (*p* = 0.02) (Appendix A). However, in most cases, the mMKM max dose was situated inside the HD-PTV. Overall, D_RBE|LEM-I_, D_RBE|mMKM_ statistics (D98%, D95%, D50%, and D2%) for distally patched plans for HD-CTV (Figure 3), and for HD-PTV and GTV (Appendix A) were in the acceptable range (±3%, compared to unpatched plans). The D_RBE|LEM-I_ conformity index for HD-CTV in large tumors with distally patched plans is 0.95 ± 0.04 compared to 0.96 ± 0.03 for unpatched plans, and for D_RBE|mMKM_, the conformity index decreases from 0.91 ± 0.08 for unpatched plans to 0.88 ± 0.3 for distally patched plans (Appendix A). By nature of distal-patching beam design, homogeneity of the plan is slightly compromised, without statistically significant difference (Appendix A). Introduction of the distal-patching structures in conventional beam design improved LETd50% in HD-CTV by a 24% increase from 38 ± 3.4 keV/µm (unpatched) to 47 ± 8.1 keV/µm (distally patched). LETdmean in HD-CTV by 21% increased from 40.1 ± 3.5 keV/µm (unpatched) to 48.6 ± 8 keV/µm (distally patched). Furthermore, the fraction of GTV receiving >50 keV/µm improved from <10% in unpatched plans to ≥50% in distally patched plans in large tumors (Appendix A).

The distal patching method could improve the fraction of GTV, HD-CTV, and HD-PTV receiving high LETd (Figure 2l,o and Figure 3a,b). Moreover, LETd statistics in distally patched plans for large tumors (HD-CTV, HD-PTV, and GTV) were more favorable than those of unpatched plans in large and small tumors (Figure 3b, Appendix A, S2f, respectively). Overall, distal patching increased the LETdmean in HD-PTV by 7.1 ± 6.5 keV/µm, HD-CTV by 8.5 ± 7.3 keV/µm, and GTV by 10 ± 8.8 keV/µm.

On evaluating the low-LETd region of large pelvic sarcomas/chordomas after distal patching, we found that there was statistically significant improvement in LETd distribution in these regions for HD-PTV, HD-CTV, and GTV, from 32 keV/µm to 37 keV/µm (Figure 4).

This unique beam design with distal patching resulted not only in improvement in the high-LETd component inside critical targets but also resulted in redistribution of the high-LETd component from the periphery to the center of the GTV and HD-CTV (Figure 5a,b). To evaluate the magnitude of this high-LETd redistribution in the central region of the GTV, we created concentric sub-volumes by shrinking the GTV in large tumors by 0.5 cm, 1.0 cm, 1.5 cm, and 2.0 cm shells. Then we compared LVHs in these concentric volumes in distally patched and unpatched plans in large tumors. A newly created sub-volume of GTV−1.5 cm shell for large tumors had comparable volume to GTV in small tumors. On comparing the LETd profiles amongst concentric sub-volumes of the GTV−1.5 cm shell in distally patched plans, unpatched plans in large tumors demonstrated very significant improvement in the high-LETd component after patching. Additionally, the LETd distribution in the GTV−1.5 cm shell in distally patched plans was also superior to the LETd distribution in the GTV of small tumors (Figure 5c). A more favorable redistribution of the high-LETd component was observed in cases where distal patching was applied to 3-beam arrangements (2 lateral and one vertical beam) than to those with 2 orthogonal beams. Such LETd redistribution is especially clinically interesting, as most often, the central portions of sarcomas/chordomas harbor radioresistant, hypoxic cells. This radioresistant, hypoxic portion in large pelvic sarcomas/chordomas can be potentially targeted by such LETd optimization methods.

### 3.4. Evaluation of D_RBE|LEM-I_, D_RBE|mMKM_ and LETd Distributions for OARs in LETd Optimization by ‘Distal Patching’

Outside the target volume, the effect of high LETd on OARs cannot be separated by the D_RBE_ in the same voxels. Hence, while evaluating impact of LETd on critical OARs such as the rectum and small intestines, we filtered out D_RBE_ < 10% prescription dose as if these organs received very low dose; then even very high LETd exposure may not be able to cause severe toxicity. We observed no significant difference in doses to OARs for both D_RBE|LEM-I_, D_RBE|mMKM_ when distally patched plans for large tumors were compared with unpatched plans for large and small tumors. LVHs and DVHs were also similar for urinary bladder, cauda equina, sacral nerve roots, and skin for distally patched and unpatched plans (Appendix A). In rectosigmoid, small intestines/bowel loops, and a region of a shell of 1 cm around the LD-PTV, LETd was lower in distally patched plans compared with unpatched plans in both the groups (Appendix A).

We also evaluated the impact of distal patching on D_RBE|LEM-I_, D_RBE|mMKM_ distribution in the entrance dose region. For this, we created a 5 cm shell beyond LD-PTV as an entrance dose region and we analysed the difference in irradiated volumes at isodose-surfaces 5, 10, 20, 30, 40, and 50 Gy (RBE) between unpatched and distally patched plans. We observed no statistical difference between the LEM-I and mMKM DVHs of the entrance dose region (5 cm shell). Also, there was no difference in the volumes of isodose surfaces between 5–50 Gy (RBE) in the entrance dose region between distally patched and unpatched plans (Appendix A). Additionally, dose distribution in the entrance dose region was assessed by means of a line-dose plot along the midplane of the HD-PTV in the antero-posterior and lateral directions. Line-dose plots demonstrated minimal differences between distally patched and unpatched plans, as depicted in a representative case (Appendix A).

We also evaluated the robustness of distally patched plans in one representative case (Appendix A), with respect to range (density ±3.5%) and setup uncertainties [±5 mm shifts in antero-posterior (AP), supero-inferior (SI), and right-left (RL)] for D_RBE|LEM-I_, D_RBE|mMKM_ distribution. Distally patched plans showed a slight compromise in LEM-I target coverage for HD-CTV for antero-posterior shifts and an increase in mMKM max doses for HD-CTV for antero-posterior shifts and range uncertainties (Appendix A).

Overall, a statistically significant difference was noted in the LETd distribution between large and small tumors, especially for 56 cm^3^, 27 cm^3^, and 9 cm^3^ (low-LETd regions) of HD-PTV, HD-CTV, and GTV, respectively. To improve LETd distribution in large tumors, LETd optimization using ‘distal patching’ was explored as a treatment-planning exercise (patients were not treated with LETd optimized plans). Distal patching for large pelvic sarcomas/chordomas resulted in an increase in LETd50% in the HD-CTV, GTV fraction receiving LETd of ≥50 keV/µm, and LETdmin in the low-LETd region of the HD-CTV, along with favorable spatial redistribution of the high-LETd component.

## 4. Discussion

Management of large unresectable pelvic sarcomas/chordomas is challenging even with CIRT, owing to radioresistant hypoxic regions in the center of tumors, tumor heterogeneity, or the presence of cancer stem-like cells. Hence, the D_RBE_ prescription cannot always translate into 100% tumor kill effect. Published studies on CIRT for unresectable sarcomas/chordomas reported that the tumor control is directly related to tumor size. Many also reported a cutoff of 200–500 cm^3^ volume for various target volumes (GTV, CTV, or PTV) as a defining factor [1,2,3,4,5,6,7]. But these target volumes can be highly variable across different centers, as the definition of target volumes, especially elective CTV (LD-CTV) and,, consequently PTV, changes with respect to contouring guidelines. In order to maintain consistent threshold criteria, we adopted HD-CTV ≥ 250 cm^3^ as a threshold criterion for consistently defining large tumors. While we followed LEM-I-based optimization for all CIRT plans, recomputation and critical evaluation of D_RBE|mMKM_ was also followed to ensure optimal coverage of HD-CTV. In very critical cases, reoptimization CIRT plans was triggered based on insufficient D_RBE|mMKM_ coverage. This was based on the report by Molineli et al. [32], where a significant undercoverage with mMKM doses was observed when plans were optimized with the LEM-I model for sacral chordomas. Despite uncertainties in the recalculation in two RBE models [45], unlike previous reports, our strategy enabled us to achieve at least HD-CTV, D_RBE|95%_ > 95% of prescription dose in both LEM-I and mMKM systems. Hence, we adopted a bi-model (LEM-I optimization and mMKM and LEM-I evaluation) approach for CIRT-plan optimization and assessment. However, a study by Matsumoto et al. [30] confirmed that even CIRT plan optimization by models other than LEM-I do not completely describe variation in RBE and LETd distribution. Similarly, in our study also, in spite of adequate coverage in both D_RBE_ models, we noticed a remarkable difference in LETd distribution between small and large pelvic chordomas/sarcomas with a significantly lower high-LETd component, especially in the central portions of targets in large tumors. The above findings are in line with reports by Matsumoto et al. and Molineli et al. [30,32], where the high-LETd component inside the target decreased as a function of tumor volumes, resulting in clinical relapses. This difference in LETd distribution can be explained: as the depth of the tumor increases along the beam direction (longer is the SOBP extension), the low-LETd component in the proximal/central-portion of target increases and the high LETd component is limited to the distal portion of the PTV and beyond the PTV. This LETd deficiency, specifically in the proximal or central portion of large tumors, can be explained by the LETd dilution through secondary low-LETd fragments generated through in-flight nuclear reactions with increasing depth [46].

The above findings suggest that apart from several clinical factors that can cause such relapses, lower intratumoral LETd in significant portions of tumors could be one of the contributing factors for poorer outcomes in large pelvic sarcomas/chordomas, in spite of fulfilling D_RBE|LEM-I_, D_RBE|mMKM_ prescriptions. Hence, in the recent past, there has been growing interest in evaluating various LET parameters that may predict relapse in such large radioresistant tumors [30,31,32]. However, there is no consensus on which LETd parameter can predict poor outcome in the best possible way and should be optimized to improve outcomes. Hagiwara et al. [31] reported that pancreatic cancer cases with intratumoral LETdmin of ≥44 keV/μm inside the GTV had significantly higher 18-month local control rates of 100.0% compared with 34.3% in those without. Matsumoto et al. [30] observed a correlation between the fraction of the tumor volume receiving <50 keV/μm and local relapse in unresectable chondrosarcomas. Molineli et al. [32] reported a median target LETd in relapsed cases of 27 keV/μm vs. 30 keV/μm in those which were controlled. Owing to the large variation in the target volumes in our cohort, rather than finding out LETdmin, LETdmean, or LETdmax in tumors, we concentrated on the critical absolute-volume of the target receiving the lowest LETd i.e., the low-LETd region that may determine the recurrence if not treated with adequately high LETd. Based on the results of this analysis, we concluded that to convert the LETd behaviour of large tumors similar to that of small tumors, one needs to shift the absolute-volume LVH curve for HD-CTV in large tumors rightwards apart from maintaining adequate D_RBE|LEM-I_, D_RBE|mMKM_ prescription dose coverage. This low-LETd region of the target (27 cm^3^ for HD-CTV, 56 cm^3^ for HD-PTV, and 9 cm^3^ for GTV) must be treated with adequately high LETd (≥33–40 keV/µm) so that the large tumors behave like small tumors in terms of LETd when adequately covered by D_RBE_ prescriptions. Another use of LETd evaluation was highlighted by Moreli et al. [47], as a prognostic factor in sacral chordomas.

The next step was to investigate solutions to overcome this intrinsic LETd deficiency with carbon ions for large tumors. For this purpose, in this study we introduced LETd optimization using ‘distal patching’ in beam designs to stop beams 1–2 cm beyond the midplane of the HD-PTV. Bassler et al. [46] also proposed a similar method for LETd painting in a phantom study where four perpendicular CIRT beams were used to treat cubical geometry, and the depth of SOBP stopped halfway through the target so that each individual beam treated only the proximal half of the target. This led to displacement of the high-LETd region from the periphery of the target (distal end of the beam path) to the center of the cubical target and improved the LETd inside the target. Similarly, in our study, simply introducing distal-patching structures in conventional beam design improved the median LETd inside the targets in distally patched plans for large tumors more than for unpatched plans for large and small tumors. Additionally, the fraction of tumor (GTV) receiving ≥50 keV/µm increased from <10% to >50%. This was a promising development, considering Matsumoto et al. reported no recurrences in unresectable pelvic chondrosarcomas if the fraction of tumor volume receiving LETd < 50 keV/µm is restricted to ≤44%. Furthermore, in our cohort, evaluation of the low-LETd region showed significant improvement in LETd distribution in distally patched plans of large tumors.

In addition to overall high-LETd distribution inside the tumor, a favorable spatial distribution of the high-LETd component was observed from the periphery of the tumor to the center (GTV−1.5 cm shell) in distally patched plans of large tumors. This spatial redistribution of the high-LETd component to the center of the tumor can be exploited for hypoxia targeting at the central portion of large sarcomas/chordomas, which often bear hypoxic, radioresistant cells. These findings were earlier reported by Hagiwara et al., where higher relapses were observed in pancreatic cancers showing low-LETd regions in the radioresistant hypoxic center of the tumor [31]. Furusawa et al. [48] confirmed that the high LETd of carbon ions ≥50 keV/µm significantly decreases the oxygen enhancement ratio (OER) and offers efficient hypoxia targeting. Similar hypoxia targeting by LETd painting by charged particles was proposed by Tinganelli et al. [49]. Bassler et al. [46,50] also proposed the feasibility of hypoxia targeting by LETd painting using different beam designs. In our findings, the redistribution of high-LETd iso-surfaces, especially 50 keV/µm to the center of the tumor by distal patching (present study) or high-LETd painting (after availability of LETd optimization tools in clinical TPS), or optimization of the high-LETd component filtered physical dose optimization [43], apart from D_RBE_, may help in bridging the gap between the dose prescribed and the optimal tumor kill-effect achieved.

Some interesting projects are being carried out by NIRS/QST, Japan and Heidelberg Ion therapy Center, Germany, where the use of multiple ions such as carbon, oxygen, and helium ions for LETd painting to improve outcomes in radioresistant tumors is being evaluated [33,51,52,53]. Konho et al. recently published on LET painting (LETd optimization functionality was available in the in-house treatment planning system) in patients with head and neck cancer treated with CIRT at NIRS/QST, Japan. They observed significant improvement in LETdmin with LET painting when compared with conventional intensity-modulated carbon-ion therapy (IMIT). In their cohort, LETd painting improved LETd in GTV by 8 to 24 keV/µm compared with conventional IMIT. [34]. In our study also, distal patching improved LETdmean in GTV by 10 ± 8.8 keV/µm. However, our patient cohort had much larger tumor volumes than those of the head and neck cancer patients reported in above study. Moreover, at the time of inception of this study, both LETd painting optimization of the high-LETd component and filtered physical dose optimization were either not commercially available at all or had a very limited availability. Additionally, a multi-ion optimization solution is currently limited to very few centers, whereas beam modifications like ‘distal patching’ can be easily employed in clinical TPS.

Another positive impact of spatial redistribution of the high-LETd component from the periphery (at the end of beam range) to the center of a target is that increasing the high-LETd component in the target showed a positive protective impact on OARs, in particular rectosigmoid, small intestines/bowel loops, and a region of a shell of 1 cm around the LD-PTV. This effect could be attributed to the redistribution of the high-LETd component from the periphery to the center in distal patching. As a matter of fact, we believe that high LETd on OARs can only be lethal when it is combined with a moderate-to-high dose in the same region of the OARs, which can significantly compromise the tolerance of OARs. No difference was noted in terms of the entrance region doses in unpatched and distally patched plans.

Obviously, the beam design modification by distal patching did not come without drawbacks. We observed max doses of up to 110% of mMKM prescription doses in distally patched plans; however, they were always situated inside the HD-CTV. Additionally, distal patching results in sharper gradients of physical dose with potential concerns on plan robustness with respect to range and setup uncertainties. In order to minimize this issue, in the present study, only 6–7 fractions (cone down boost treatment to HD-PTV) were planned with distal patching. Optimizing overlap between distally patched beams may improve the robustness of these plans. The comprehensive evaluation of issues related to robustness of distally patched plans with respect to range and setup errors was beyond the scope of this paper.

Another limitation of this study is that this study is essentially a planning study. We did not focus on clinical outcomes such as local recurrence, as comprehensive evaluation of treatment response and assessment of their relationship with various LETd parameters was limited by the short follow-up available in this study. Additionally, establishing correlation between LETd parameters and local recurrences is a complex issue. The antitumor effect of CIRT involves not only local cell killing but also cell signaling pathways, metabolic pathways (cellular antioxidant capacity), abscopal effects, the bystander effect, and the anti-tumor immune mechanism [19,54,55,56,57]. Low LETd irradiation with CIRT may risk both the local and global failure of tumors involving the above-mentioned mechanism. Moreover, it is extremely difficult to determine which specific region of the tumor the recurrence originated from. Hence, correlating the origin of recurrence with low-LETd or high-LETd regions is extremely difficult.

Nevertheless, the strength of our approach is that we have independently derived LETd parameters based on small and large tumor geometries, focusing on the hypothesis that low LETd could be one of the contributing factors for development of local recurrences in large pelvic sarcomas/chordomas. There is very limited literature available on LETd optimization in CIRT. To our knowledge, this is the first study evaluating the feasibility of LETd optimization in large pelvic sarcomas/chordomas. And despite the small number of patients, this study is a promising step in the direction of exploring and introducing simple and clinically achievable LET optimization methods. The assessed LETd parameters in this study can be considered as a starting point for future LETd optimization. In the future, we intend to collaborate with multiple institutions with larger databases and with longer follow-ups and explore in an independent population whether our LETd parameters are predictive of recurrence. Such future studies may actually enable us to determine if low intratumoral LETd is indeed one of the driving factors for local recurrences in large pelvic sarcomas/chordomas.

## 5. Conclusions

In this study, a statistically significant difference was observed between small and large tumors in terms of LETd distribution, despite comparable D_RBE_ distribution. LETd optimization using distal patching significantly improved LETd distribution in tumors without significantly compromising the D_RBE_ profile in the target or compromising OAR constraints. Spatial redistribution of high LETd by distal patching in the center of the GTV may offer the possibility to overcome a radioresistant/hypoxic component. There is a tradeoff between an optimal D_RBE_/LETd profile with LETd optimization using the distal-patching method and robustness against setup and range uncertainties. Detailed evaluation on robustness with distal patching is required. However, initial results of our analysis are promising. Whether the low-LETd region in the tumor might be responsible for recurrence is still a hypothesis. In future studies, we intend to confirm this in a larger dataset by involving other centers and we plan to confirm this in a prospective study before implementing it in routine clinical practice.

## Figures and Tables

**Figure 1 cancers-15-04903-f001:**
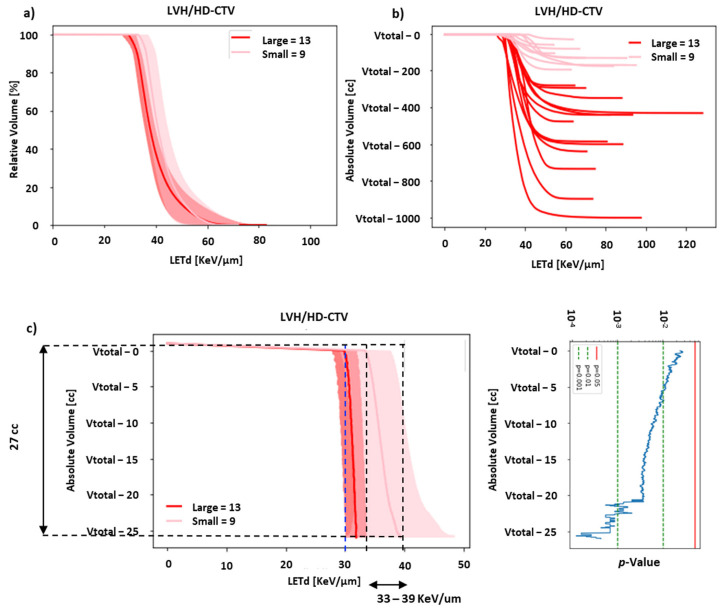
LVH evaluation for HD-CTV in small and large pelvic sarcomas/chordomas. (**a**) A relative-volume cumulative LVH for HD-CTV of small and large pelvic sarcomas (mean ± SD), showing difference in LETd distribution. (**b**) An absolute-volume LVH of individual cases shifted with respect to largest HD-CTV (Vtotal = volume of largest HD-CTV), demonstrating distinct separation between LVH plots of large and small tumors. (**c**) An absolute-volume cumulative LVH for HD-CTV of small and large pelvic sarcomas (mean ± SD), showing a statistically significant difference (*p* < 0.05) for 27 cm^3^ of HD-CTV (low-LETd region) that should receive LETd of at least ≥33–39 keV/µm. Note: in absolute LVH in (**c**), the blue dashed line represents average LETdmin in HD-CTV of large tumors for unpatched CIRT plans. In order to improve LETd distribution in large tumors the LVH curve must be shifted to right (towards black dashed line i.e., average LETdmin ± SD in HD-CTV of small tumors). Green dashed line and red line in *p*-value plot represents line *p*-value cutoffs of 0.001, 0.01 and 0.05 respectively.

**Figure 2 cancers-15-04903-f002:**
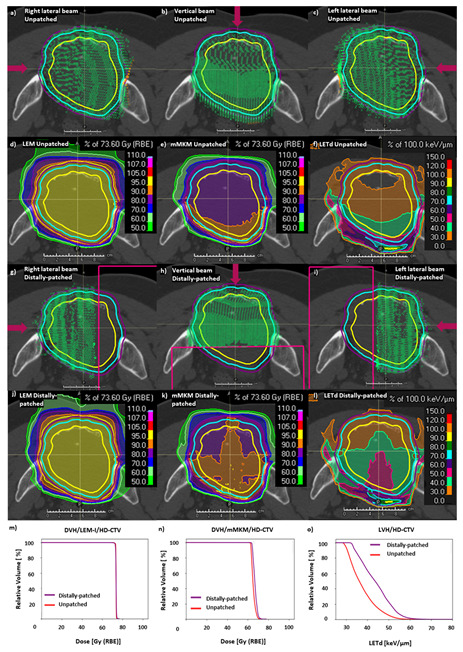
LETd optimization in large pelvic sarcomas/chordomas with LETd optimization using ‘distal patching’. (**a**–**c**) Description of unpatched beam design, a conventional 3-field beam design (bilateral and posterior beam) in a representative case of large pelvic sarcoma with spatial distribution of carbon-ion spots (green crosses represent carbon-ion spots, purple arrows represent direction of the beams). (**d**) D_RBE|LEM-I_, (**e**) D_RBE|mMKM_, and (**f**) LETd (unpatched) distribution corresponding to Figure 1a–c. The GTV (yellow), HD-CTV (cyan), HD-PTV (purple) were prescribed 73.6 Gy (RBE)/16 fraction. The high-LETd region can be seen at the end of the carbon-ion beam range in unpatched-CIRT plan (**f**). (**g**–**i**) Description of a “distal patching” with spot distribution in 3-field beam design (bilateral and posterior beam) with introduction of distal-patching structures, applied to cone down boost to HD-PTV (7 fractions). (**j**–**l**) Images demonstrating the impact of distal patching on the D_RBE|LEM-I_, D_RBE|mMKM_ distribution and redistribution of LETd (especially, 40 and 50 keV/µm LETd surface in green and magenta, respectively) from periphery of HD-CTV and HD-PTV (**f**) to the central region of GTV (**l**). (**m**–**o**) DVH and LVH comparison between non-distally patched (red), and distally patched (purple) CIRT plans. Slightly higher mMKM max doses were observed, but they were located inside the target in the distally patched plan. (**o**) LVH showing significant improvement on LETd distribution in the distally patched CIRT plans. Note: distally-patched data (**j**–**o**) represent statistics from unpatched plan for 9 fractions, and from distally-patched plan for 7 fractions.

**Figure 3 cancers-15-04903-f003:**
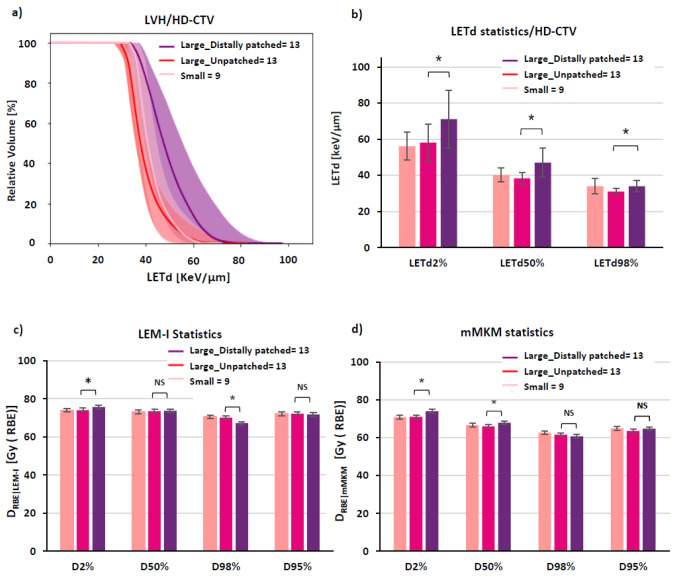
LETd optimization using ‘distal patching’: D_RBE|LEM-I_, D_RBE|mMKM_, and LETd statistics and LVH for HD-CTV. (**a**) Cumulative relative-volume LVH for HD-CTV (mean ± SD) displaying improvement in LETd distribution in large tumors after distal patching. (**b**) LETd statistics displaying statistically significant improvement in all the LETd parameters, especially LETd98%, where major part of HD-CTV in large tumors receives LETd comparable to small tumors. (**c**,**d**) LEM-I and mMKM statistics showing comparable target coverage in small tumors (unpatched), large tumors (unpatched) and large tumors with distally patched plans. Note: statistics is displayed for 10–9 fractions of unpatched + 6–7 fractions of distally patched CIRT plan). *: *p* < 0.05, NS: *p* = not significant.

**Figure 4 cancers-15-04903-f004:**
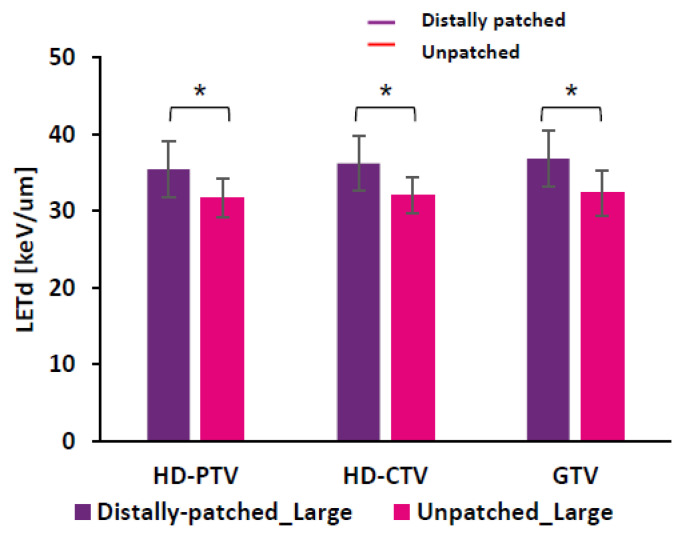
LETd distribution in low-LETd region of HD-PTV improved from 31.7 ± 2.5 keV/µm to 35.4 ± 3.6 keV/µm, for HD-CTV improved from 32 ± 2.3 keV/µm to 36.2 ± 3.6 keV/µm, and for GTV improved from 32.3 ± 3 keV/µm to 36.8 ± 3.7 keV/µm. (* *p* < 0.05).

**Figure 5 cancers-15-04903-f005:**
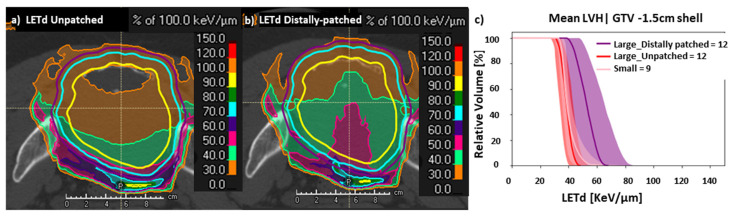
LETd optimization ‘distal patching’: Spatial redistribution of high-LETd component in the central region of the targets. (**a**) Redistribution of LETd distribution was observed from periphery of HD-PTV (purple) and HD-CTV (cyan) in unpatched plans (**b**) to the central region of GTV (yellow) and HD-CTV in distally patched plans [sub-volumes created by shrinking the GTV in large tumors by 0.5 cm (blue), 1.0 cm (orange), 1.5 cm (red)]. (**c**) LVH comparison between central region of GTV in large tumors with distally patched and unpatched plans with comparable volumes of GTV in cases with small tumor. The concentric sub-volume of GTV−1.5 cm shell in distally patched plans in large tumors showed significant improvement in LETd distribution, with significantly higher LETd50%.

**Table 1 cancers-15-04903-t001:** Patient/Tumor Characteristics.

Patient Characteristics		Small	Large	*p*-Value
		(n = 9)	(n = 13)	Small vs. Large
Age	Median [years]	64	63	NS *
	Range [years]	42–76	43–77	
Gender	Male	7	12	NS
	Female	2	1	NS
Follow-up	Median [months]	15	14	NS
	Range [months]	4–23	4–28	
Histology	Chordoma	8	13	NS
	Sarcoma (Synovial)	1	0	NS
Surgery		1	1	NS
Chemotherapy		0	1	NS
Tumor Characteristics				
GTV	Mean ± SD [cm^3^]	55.9 ± 39.8	301 ± 243.5	0.004
HD-CTV	Mean ± SD [cm^3^]	116.3 ± 52.6	551.7 ± 211.3	<0.001
HD-PTV	Mean ± SD [cm^3^]	195.1 ± 76.8	776.7 ± 257.7	<0.001
Maximum GTV diameter along the beam path	[cm]	5.4 ± 2.1	9.1 ± 3.8	0.01
CIRT dose [LEM-I]	Median [Gy (RBE)]	73.6	73.6	NS
	Range [Gy (RBE)]	70.4–73.6	70.4–73.6	
D_RBE_ Statistics				
GTV	LEM-I			
	D2% Mean ± SD [Gy (RBE)]	73.7 ± 1.6	74.5 ± 0.6	NS
	D50% Mean ± SD [Gy (RBE)]	73.1 ± 1.3	73.8 ± 0.2	NS
	D98% Mean ± SD [Gy (RBE)]	71.5 ± 1.1	72.0 ± 1.9	NS
	mMKM			
	D2% Mean ± SD [Gy (RBE)]	69.0 ± 2.9	71.6 ± 3.8	NS
	D50% Mean ± SD [Gy (RBE)]	65.4 ± 2.1	66.6 ± 2.5	NS
	D98% Mean ± SD [Gy (RBE)]	63.1 ± 2.5	63.7 ± 1.3	NS
HD-CTV	LEM-I			
	D2% Mean ± SD [Gy (RBE)]	74.1 ± 1.4	74.1 ± 1.0	NS
	D50% Mean ± SD [Gy (RBE)]	73.3 ± 1.0	73.5 ± 0.9	NS
	D98% Mean ± SD [Gy (RBE)]	70.5 ± 2.8	70.1 ± 2.8	NS
	mMKM			
	D2% Mean ± SD [Gy (RBE)]	70.9 ± 1.8	71.0 ± 3.6	NS
	D50% Mean ± SD [Gy (RBE)]	66.6 ± 1.4	66.1 ± 2.7	NS
	D98% Mean ± SD [Gy (RBE)]	62.7 ± 3.8	61.6 ± 2.2	NS
HD-PTV	LEM-I			
	D2% Mean ± SD [Gy (RBE)]	74.1 ± 1.3	74.1 ± 0.9	NS
	D50% Mean ± SD [Gy (RBE)]	73.2 ± 1.0	73.4 ± 0.9	NS
	D98% Mean ± SD [Gy (RBE)]	65.0 ± 10.2	66.5 ± 5.6	NS
	mMKM			
	D2% Mean ± SD [Gy (RBE)]	72.6 ± 1.7	72.4 ± 3.2	NS
	D50% Mean ± SD [Gy (RBE)]	67.01 ± 1.6	66.3 ± 2.6	NS
	D98% Mean ± SD [Gy (RBE)]	56.9 ± 12.4	56.3 ± 7.0	NS
LETd Statistics				
GTV	LETd2% Mean ± SD [KeV/µm]	51.1 ± 9.6	55.5 ± 9.8	NS
	LETd50% Mean ± SD [KeV/µm]	37.6 ± 4.0	37.2 ± 2.2	NS
	LETd98% Mean ± SD [KeV/µm]	33.6 ± 3.9	31.2 ± 2.0	NS
HD-CTV	LETd2% Mean ± SD [KeV/µm]	56.1 ± 7.7	58.1 ± 10.1	NS
	LETd50% Mean ± SD [KeV/µm]	40.2 ± 3.8	38.3 ± 3.2	NS
	LETd98% Mean ± SD [KeV/µm]	34.0 ± 4.4	31.1 ± 1.7	NS
HD-PTV	LETd2% Mean ± SD [KeV/µm]	67.4 ± 18.0	69.6 ± 12.5	NS
	LETd50% Mean ± SD [KeV/µm]	45.0 ± 4.9	39.9 ± 3.2	0.02
	LETd98% Mean ± SD [KeV/µm]	40.3 ± 10.5	31.1 ± 2.0	0.04

* NS: not significant.

## Data Availability

The data presented in the current study are available from the corresponding author (A.N.) upon reasonable request.

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
