# Peer review of "Planning Strategy to Optimize the Dose-Averaged LET Distribution in Large Pelvic Sarcomas/Chordomas Treated with Carbon-Ion Radiotherapy"

_cancers, 2023, doi:10.3390/cancers15194903_

Round 1
Reviewer 1 Report
1-Please identify the novelty of this work and its necessity.
2-In introduction section, more literature are needed for Carbon-ion therapy.
3-There is no any information about the Radiotherapy Machine, its location and also the software of treatment planning.
4- More explanation about the carbon-ion as a treatment probe is necessary.
Minor editing of English language and typo error are required.
Author Response
We sincerely thank Reviewer #1 for reviewing our work and providing constructive feedback. We tried to address all the issues raised by Reviewer #1 in following format and the changes made in the manuscript are highlighted in red. Wherever relevant we highlighted original text in green to bring to the attention of reviewers.
The detailed response is attached in the file

Reviewer 2 Report
The article "Planning strategy to optimize the dose averaged LET distribution in large pelvic sarcomas chordomas treated with carbon-ion radiotherapy" by Ankita Nachankar et al. is in my opinion clear and comprehensive. The work is easily understood even by non-experts in the subject and is in a context where the literature is rather scarce due to the limited number of centers practicing this type of therapy. In my opinion, it can be published in the present form.

Author Response
We sincerely thank Reviewer #2 for reviewing our work and providing constructive feedback. We tried to address all the issues raised by Reviewer #2 in following format and the changes made in the manuscript are highlighted in red. Wherever relevant we highlighted original text in green to bring to the attention of reviewers.

Reviewer 3 Report
In the manuscript “Planning strategy to optimize the dose averaged LET distribu-2 tion in large pelvic sarcomas/chordomas treated with carbon-ion 3 radiotherapy”, the authors aimed to evaluate 22 pelvic sarcomas/chordomas patients treated with 28 CIRT [large: HD-CTV≥250 cm3 (n = 9), small: HD-CTV<250 cm3 (n = 13)], DRBE|LEM-I = 73.6 (70.4-73.6) 29 Gy (RBE)/16 fractions, using the local effect model-I (LEM-I) optimization and modified-microdo-30 simetric kinetic model (mMKM) recomputation. Distal-patching is clinically 39 implementable without significant DRBE compromise for targets/OARs. The study look fine (looks technically sound) but some following issued need be addressed .
Major concern
1 First of all, the sample size is too small which included 22 patients ,it is difficult to validate the conclusion.
2 The title is focus on planning strategy to optimize the dose averaged LET distribution, however in main text ,you did not show the side effect of patients in specific dose in one section.
3: You divide large and small tumor group, what tumor size did you distinguish from two group? In addition ,if you can present the survival or curative effect of patients after this optimization dose planning, it will strength the clinical value of the article.
4: What is the research finding of this article? Please summarize in simple summary section.
Minor concern
1 The abbreviation need show full name when first in text ,eg RBE in simple summary section ,etc .
2 the study limitation need be added in discussion .
Author Response
We sincerely thank Reviewer #3 for reviewing our work and providing constructive feedback. We tried to address all the issues raised by Reviewer #3 in following format and the changes made in the manuscript are highlighted in red. Wherever relevant we highlighted original text in green to bring to the attention of reviewers.

Round 2
Reviewer 3 Report
accept